# An Efficient Two-Stage Receiver Base on AOR Iterative Algorithm and Chebyshev Acceleration for Uplink Multiuser Massive-MIMO OFDM Systems

Yung-Ping Tu [1,*], Chih-Yung Chen [1] and Kuang-Hao Lin [2,*]

1    Department of Electronic Engineering, National Formosa University, Yunlin 632301, Taiwan;
     10960102@gm.nfu.edu.tw
2    Department of Electrical Engineering, National Formosa University, Yunlin 632301, Taiwan
*    Correspondence: duhyp@gs.nfu.edu.tw (Y.P.-T.); khlin@nfu.edu.tw (K.-H.L.)

**Abstract:** The massive multiple-input multiple-output systems (M-MIMO) and orthogonal frequency-division multiplexing (OFDM) are considered to be some of the most promising key techniques in the emerging 5G and advanced wireless communication systems nowadays. Not only are the benefits of applying M-MIMO and OFDM for broadband communication well known, but using them for the application of the Internet of Things (IoT) requires a large amount of wireless transmission, which is a developing topic. However, its high complexity becomes a problem when there are numerous antennas. In this paper, we provide an effective two-stage multiuser detector (MUD) with the assistance of the accelerated over-relaxation (AOR) iterative algorithm and Chebyshev acceleration for the uplink of M-MIMO OFDM systems to achieve a better balance between bit error rate (BER) performance and computational complexity. The first stage of the receiver consists of an accelerated over-relaxation (AOR)-based estimator and is intended to yield a rough initial estimate of the relaxation factor $\omega$, the acceleration parameter $\gamma$, and transmitted symbols. In the second stage, the Chebyshev acceleration method is used for detection, and a more precise signal is produced through efficient iterative estimation. Additionally, we call this proposed scheme Chebyshev-accelerated over-relaxation (CAOR) detection. Conducted simulations show that the developed receiver, with a modest computational load, can provide superior performance compared with previous works, especially in the MU M-MIMO uplink environments.

**Keywords:** multiuser massive MIMO (MU M-MIMO); Internet of Things (IoT); 5G (fifth generation); orthogonal frequency-division multiplexing (OFDM); accelerated over-relaxation; Chebyshev acceleration; Chebyshev-accelerated over-relaxation (CAOR) detection



## 1. Introduction

Nowadays, the Internet of Things (IoT) of 5G [1–3] is one of the crucial applications of wireless communication systems [4–6]. Namely, wireless transmission services have become an extremely important method of Internet of Things (IoT) information delivery. The applications of the Internet of Things (IoT) include smart homes, healthcare, traffic auxiliary management, industrial automation, and crisis response for natural and artificial disaster prevention. To meet various types of applications of the Internet of Things (IoT), wireless transmission of large amounts of results to data collection centers is an essential need, i.e., 5G technology will be forced to bear massive amounts of data while being more time saving than 4G. Furthermore, the multiple-input multiple-output (MIMO) and the orthogonal frequency-division multiplexing (OFDM) are indispensable technologies of wireless communication systems currently [7,8]. The former, with multiple antennas at the transmitters and receivers, which can significantly increase the data throughput of the system and transmission distance without increasing the total transmit power expenditure or bandwidth demand, also can efficiently obtain diversity gain, array gain, capacity gain, and beamforming gain [9–11]; the latter is a favored modulation and transmission

scheme that cuts a high-data-rate stream into some lower-rate streams simultaneously transmitted over some narrow-band channels parallelly, which has a lot of advantages, such as robustness against narrow-band co-channel interference, inter-symbol interference (ISI) caused by multipath propagation, and low sensitivity to time synchronization errors [12,13]. To further promote the merits of the MIMO systems and meet higher spectrum efficiency, the pioneer researchers proposed the massive MIMO (M-MIMO) structure [14–17], which utilizes spatial diversity to support more users by mounting a large number of antennas in the base station (BS) [18–20].

It is a pity that, although the M-MIMO systems have brought many benefits, complexity is also appreciably increased. Especially as far as the receiver is concerned, conventional linear detection methods such as zero-forcing (ZF) and minimum mean-square error (MMSE) are adopted [21,22], even if they show good bit error rate (BER) performance; however, as mentioned by Yin et al. [23], since they involve matrix inversion operations of $\mathcal{O}(N^3)$ complexity, the detection complexity is too high to implement when $N$ is increasing, where $N$ is the antenna number.

Therefore, many iterative methods aim to eschew matrix inversion and not calculate it. In addition, a comparative study of low-complexity linear detectors of M-MIMO is addressed in [24] to discuss the pros and cons of iterative matrix inversion methods. To maintain good performance and low complexity, low-complexity signal detection for the uplink of massive MIMO systems is proposed, such as the Jacobi method [25], Gauss–Seidel method [26], and successive over-relaxation (SOR) method [27] proposed by Kong et al., Wu et al., and Gao et al., respectively, which have lower complexity as $\mathcal{O}(N^2)$ through the iterative procedure to avoid the notorious inverse matrix operation. However, the performance is not sufficient to satisfy us. In light of this, Yu et al. [28] considered a modified SOR method to reduce the complexity, but it is strongly confined to lower modulation orders; namely, its performance degrades substantially when the high modulation order becomes more required. In [29], Ning et al. proposed the symmetric successive over-relaxation (SSOR) method, which combines two SOR sweeps in such a way that the resulting iteration matrix is similar to a symmetric matrix and its performance is better than the previous SOR-based method; nevertheless, its complexity will increase drastically compared to the SOR-based method. Compared to the SOR-based method [27], Zhang et al. proposed the AOR method in [30], which is an extension of the SOR by using two parameters to speed up convergence, though the complexity of AOR is slightly higher than SOR; however, its performance can be significantly improved. To obtain a reasonable computational load, the Chebyshev acceleration uses the extension of the extrapolation acceleration method to accelerate convergence and reduce complexity [31].

In light of the above, to achieve a good trade-off between BER performance and complexity load, we propose a two-stage receiver for the MU M-MIMO systems, called Chebyshev-accelerated over-relaxation (CAOR) detection. The first stage of the receiver consists of setting each parameter of the AOR method [30] and using its algorithm to produce the rough initial estimate outputs, and then submitting those initial estimates of symbol detection and the AOR parameters to the second stage. In the second stage, the Chebyshev acceleration is conducted successively using the recursive relationship of the Chebyshev polynomials. It is noteworthy that, to render the estimated parameters more thoroughly, unlike the conventional iterative algorithm, this method provides Chebyshev algorithm calculation through AOR's initial estimated output and parameters to produce more precise and fast signal detection. Conducted simulations show that the proposed two-stage detection scheme, with a modest computational load, can provide superior performance compared with most iterative detection methods, especially in uplink MU M-MIMO scenarios.

The rest of this paper is organized as follows. Section 2 introduces the system model employed in this paper. A review of some conventional iterative schemes and a new two-stage receiver based on the AOR iterative algorithm and Chebyshev acceleration are addressed in Section 3. Simulation results and computational complexity analysis are

provided in Section 4 to verify the proposed approach. Section 5 provides the concluding remarks to summarize the paper.

## 2. System Model

### 2.1. Uplink Multiuser MIMO OFDMA Systems

We consider the uplink multiuser MIMO orthogonal frequency-division multiple access (OFDMA) systems [32,33] with $KN_t$ user antennas, and $N_R$ antennas at the BS, where $K$ is the user number, each user is equipped with $N_t$ transmit antennas, and $N_R \gg KN_t$. Then, we use $N_T$ to represent the total number of user antennas. The transmitted signal and the received signal can be expressed as $\mathbf{s} = [s_1, s_2, \cdots, s_{N_T}]^T$ and $\mathbf{y} = [y_1, y_2, \cdots, y_{N_R}]^T$, respectively. In addition, we use bold and capitals to denote vector and matrix, respectively. Hence, the system model can be represented by the following Equation (1):

$$\mathbf{y} = H\mathbf{s} + \mathbf{n}, \tag{1}$$

where $H$ is the $N_R \times N_T$ complex channel matrix and $\mathbf{n}$ is the $N_R \times 1$ noise vector. Moreover, it can be expressed as

$$\begin{bmatrix} y_1 \\ \vdots \\ y_{N_R} \end{bmatrix} = \begin{bmatrix} h_{11} & \cdots & h_{1N_T} \\ \vdots & \ddots & \vdots \\ h_{N_R 1} & \cdots & h_{N_R N_T} \end{bmatrix} \begin{bmatrix} s_1 \\ \vdots \\ s_{N_T} \end{bmatrix} + \begin{bmatrix} n_1 \\ \vdots \\ n_{N_R} \end{bmatrix}. \tag{2}$$

### 2.2. Channel Model

We assume that $H \in \mathbb{C}^{N_R \times N_T}$ denotes the flat Rayleigh fading channel matrix with two-channel taps that are independent and identically distributed (i.i.d.) and obey Gaussian distribution with unit variance and zero mean, and the elements in noise vector $\mathbf{n}$ are i.i.d. and have complex Gaussian distribution $\mathcal{CN}(0, \sigma^2)$.

At the BS, the task of the detector is to estimate the transmitted signal vector $\mathbf{s}$ from the received signal vector $\mathbf{y}$. According to [20], the conventional linear MMSE detection algorithm has been proven, which is near-optimal for uplink MIMO systems. In addition, the estimate of the transmitted signal vector $\mathbf{s}$ can be expressed as

$$\mathbf{s} = (H^H H + \sigma^2 I_{N_T})^{-1} H^H \mathbf{y} = W^{-1} \mathbf{y}^{MF}, \tag{3}$$

where $\sigma^2$ denotes the variance of the noise vector, $W$ is the filtering matrix of conventional MMSE, and $\mathbf{y}^{MF}$ is the output of the matched filter. Within Equation (3), we can observe that the matrix inversion operation is contained in the conventional MMSE detection. Therefore, the computational complexity of conventional MMSE is too high to realize in the MU-MIMO systems, not to mention the massive antennas.

To estimate the flat Rayleigh fading matrix at the BS, we used a comb-type pilot structure [34], with pilot tones at periodically located subcarriers. Then, we used the least square (LS) channel estimation method [35] to find the channel estimate $\hat{H}_{LS}$ by pilot tones. The result of LS channel estimation is given by [7]

$$\hat{H}_{LS} = (X^H X)^{-1} X^H Y = X^{-1} Y, \tag{4}$$

where $X$ and $Y$ denote the transmitted signal's pilot tones and the received signal's pilot tones at the subcarriers, respectively.

## 3. Proposed Scheme

To explain our proposed method more clearly, first, we will briefly describe the conventional SOR method [27], AOR method [30], and Chebyshev acceleration method [31], sequentially, and then propose a two-stage CAOR scheme to balance the performance and complexity trade-off by performing fewer iterations to further accelerate convergence.

### 3.1. Iteration Method Review

3.1.1. Conventional SOR Method

We consider a linear system [36] whose mathematical equation can be expressed as

$$A\mathbf{x} = \mathbf{b}, \tag{5}$$

where the matrix $A$ is symmetric positive-definite, $\mathbf{x}$ is the unknown vector and we can decompose $A$ as follows:

$$A = D - L - U, \tag{6}$$

in which $D$, $-L$, and $-U$ are the diagonal, strict lower part, and strict upper part of $A$. As for the iterative equation, the Jacobi iteration [25] is given by

$$\mathbf{x}^{(k+1)} = D^{-1}(L + U)\mathbf{x}^{(k)} + D^{-1}\mathbf{b}, \tag{7}$$

Likewise, the Gauss–Seidel iteration [26] also can be expressed as

$$\mathbf{x}^{(k+1)} = (D - L)^{-1}U\mathbf{x}^{(k)} + (D - L)^{-1}\mathbf{b}. \tag{8}$$

Here, we can define a matrix $G$ as

$$G = M^{-1}N. \tag{9}$$

Therefore, for the Jacobi iteration, $M = D$ and $N = (L + U)$. Moreover, for the Gauss–Seidel iteration, $M = (D - L)$ and $N = U$. Then, Equations (7) and (8) can be rewritten as

$$\mathbf{x}^{(k+1)} = G\mathbf{x}^{(k)} + \mathbf{d}, \tag{10}$$

where $G$ is called the iteration matrix and $\mathbf{d}$ is $M^{-1}\mathbf{b}$. Based on this result, we multiply the relaxation factor $\omega$ in Equation (10), and we have

$$\omega\mathbf{x}^{(k+1)} = \omega(G\mathbf{x}^{(k)} + \mathbf{d}), \tag{11}$$

which gives the SOR iteration [27]. Take the Gauss–Seidel iteration as an example, which is defined as

$$\mathbf{x}^{(k+1)} = (D - \omega L)^{-1}\left\{[(1 - \omega)D + \omega U]\mathbf{x}^{(k)} + \omega\mathbf{b}\right\}. \tag{12}$$

According to [27], we can determine that the SOR iterative algorithm is convergent.

3.1.2. AOR Method

The accelerated over-relaxation (AOR) iterative algorithm [37] can be regarded as an extension of the successive over-relaxation (SOR) iterative algorithm, and its iterative equation is as follows:

$$
\begin{aligned}
\mathbf{x}^{(k+1)} = (D - \gamma L)^{-1}&[(1 - \omega)D + \omega U + (\omega - \gamma)L]\mathbf{x}^{(k)} \\
&+ \omega(D - \gamma L)^{-1}\mathbf{b},
\end{aligned} \tag{13}
$$

where $\omega$ and $\gamma$ are the relaxation parameter and the acceleration parameter, respectively. Furthermore, for specific values of $\omega$ and $\gamma$, when $\omega = \gamma$ or $\omega = \gamma = 1$, the AOR iterative algorithm can reduce to the SOR or the Gauss–Seidel iterative algorithm, respectively.

According to *Lemma* 1 in [27], in uplink large-scale MIMO, the conventional MMSE filtering matrix $W$ is a symmetric positive definite matrix, and it can be decomposed as

$$W = D + L + L^H. \tag{14}$$

where $D$, $L$, and $L^H$ are the diagonal, strict lower part, and strict upper part of $W$. Therefore, we can use the AOR iterative algorithm to rewrite the conventional MMSE algorithm as follows:

$$
\begin{aligned}
\mathbf{x}^{(k+1)} = (D + \gamma L)^{-1}[(1 - \omega)D - \omega L^H - (\omega - \gamma)L]\mathbf{x}^{(k)} \\
+ \omega(D + \gamma L)^{-1}\mathbf{y}^{MF},
\end{aligned}
\tag{15}
$$

and the relaxation parameter $\omega$ and the acceleration parameter $\gamma$ of the AOR iterative method are given by [37] as follows:

$$
\omega = \frac{1}{\sqrt{1 - \mu^2}},
\tag{16}
$$

$$
\gamma = \frac{2}{1 + \sqrt{1 - \mu^2}},
\tag{17}
$$

where $\mu = \rho(G_{\gamma,\omega})|_{\gamma=0,\omega=1}$; we will also define $\rho(G_{\gamma,\omega})$ in later sections.

AOR Convergence Analysis

We know that in any iterative algorithm in the form of Equation (10), when $k$ reaches infinity, $\mathbf{x}^{(k+1)}$ will be equal to $\mathbf{x}^{(k)}$. Noting Equation (15), we can use Equation (10) to define $G_{\gamma,\omega} = (D + \gamma L)^{-1}[(1 - \omega)D - \omega L^H - (\omega - \gamma)L]$ and $\mathbf{d} = \omega(D + \gamma L)^{-1}\mathbf{y}^{MF}$; afterward, Equation (15) can be rewritten as

$$
\mathbf{x}^{(k+1)} = G_{\gamma,\omega}\mathbf{x}^{(k)} + \mathbf{d},
\tag{18}
$$

and the spectral radius of $G_{\gamma,\omega}$ is defined as the non-negative number as follows:

$$
\rho(G_{\gamma,\omega}) \triangleq \max_{\lambda \in \rho(G_{\gamma,\omega})} |\lambda|,
\tag{19}
$$

in which $\lambda$ is the eigenvalue of $G_{\gamma,\omega}$. Moreover, if the spectral radius $\rho(G_{\gamma,\omega})$ satisfies

$$
\rho(G_{\gamma,\omega}) = \max_{1<n<N_T} |\lambda_n| < 1,
\tag{20}
$$

Therefore, Equation (15) will converge [31]. According to the theorem of eigenvalues, we can obtain

$$
G_{\gamma,\omega}\mathbf{x} = (D + \gamma L)^{-1}[(1 - \omega)D - \omega L^H - (\omega - \gamma)L]\mathbf{x} = \lambda_n\mathbf{x},
\tag{21}
$$

where $\mathbf{x}$ is an arbitrary $N_T \times 1$ non-zero real-valued vector and Equation (21) also can be rewritten as

$$
[(1 - \omega)D - \omega L^H - (\omega - \gamma)L]\mathbf{x} = (D + \gamma L)\lambda_n\mathbf{x}.
\tag{22}
$$

Multiplying both sides of Equation (22) by the conjugate transpose of $\mathbf{x}$, we can obtain

$$
\mathbf{x}^H[(1 - \omega)D - \omega L^H - (\omega - \gamma)L]\mathbf{x} = \mathbf{x}^H(D + \gamma L)\lambda_n\mathbf{x}.
\tag{23}
$$

Next, we take the conjugate transpose on both sides of Equation (23), in which $D$ is a diagonal matrix, so $D = D^H$, and then we can obtain an updated equation as

$$
\mathbf{x}^H[(1 - \omega)D - \omega L - (\omega - \gamma)L^H]\mathbf{x} = \mathbf{x}^H(D + \gamma L^H)\lambda_n\mathbf{x}.
\tag{24}
$$

Then, adding Equations (23) and (24) will obtain

$$
\begin{aligned}
\mathbf{x}^H[(2 - 2\omega)D - \omega(L + L^H) - (\omega - \gamma)(L + L^H)]\mathbf{x} \\
= \mathbf{x}^H\lambda_n[2D + \gamma(L + L^H)]\mathbf{x}.
\end{aligned}
\tag{25}
$$

Substituting Equation (14) into Equation (25), we can obtain

$$(\lambda_n - 1)(\gamma - 2)\mathbf{x}^H D \mathbf{x} = [\lambda_n \gamma + (2\omega - \gamma)]\mathbf{x}^H W \mathbf{x}, \tag{26}$$

Since the filter matrix $W$ of conventional MMSE is positive definite, as demonstrated in [27], $D$ is also positive definite, i.e., the diagonal matrix of $W$. Thus, we have $\mathbf{x}^H D \mathbf{x} > 0$ and $\mathbf{x}^H W \mathbf{x} > 0$. Moreover, if both $(\gamma - 2)$ and $(\lambda_n - 1)$ are less than zero, i.e., $0 < \gamma < 2$ and $\lambda_n < 1$, then

$$(\lambda_n - 1)(\gamma - 2) > 0, \tag{27}$$

and $\lambda_n \gamma + (2\omega - \gamma) > 0$. Therefore, we can determine that $(\lambda_n - 1)[\lambda_n \gamma + (2\omega - \gamma)] < 0$, that means

$$\lambda_n[(\lambda_n - 2)\gamma + 2\omega] < 2\omega - \gamma. \tag{28}$$

We know that when $\omega = \gamma$, we can obtain

$$|\lambda_n| < 1. \tag{29}$$

Paying attention to Equation (29), if we give appropriate values of $\omega$ and $\gamma$, it can be guaranteed to comply with $\rho(G_{\gamma,\omega}) < 1$. Thus, the AOR iterative algorithm is convergent.

### 3.1.3. Chebyshev Acceleration

Chebyshev acceleration [31] can be regarded as the promotion of extrapolation acceleration. Consider $\mathbf{x}^{(0)}, \mathbf{x}^{(1)}, ..., \mathbf{x}^{(k)}$, if $\varepsilon_k = \mathbf{x}^{(k)} - \mathbf{x}^*$ is defined as the error of the $k$th iteration solution, in which $\mathbf{x}^*$ is a target signal and $\mathbf{x}^{(k)}$ is $k$th-time iteration output; then, $\varepsilon_k$ can be expressed as

$$\varepsilon_k = G\varepsilon_{k-1} = G^2\varepsilon_{k-2} = \cdots = G^k\varepsilon_0, \tag{30}$$

where $G$ is the iteration matrix and $G^k$ is $k$th-time iteration matrix. Assume that $\tilde{\mathbf{x}}^{(k)}$ is the linear combination of $\mathbf{x}^{(0)}, \mathbf{x}^{(1)}, ..., \mathbf{x}^{(k)}$ as follows

$$\tilde{\mathbf{x}}^{(k)} = \alpha_0 \mathbf{x}^{(0)} + \alpha_1 \mathbf{x}^{(1)} + \cdots + \alpha_k \mathbf{x}^{(k)}, \tag{31}$$

where $\alpha_k$ is undetermined coefficients, and it satisfies $\sum_{i=0}^{k} \alpha_i = 1$. Therefore, $\tilde{\mathbf{x}}^{(k)} - \mathbf{x}^*$ can be defined as

$$\tilde{\mathbf{x}}^{(k)} - \mathbf{x}^* = \alpha_0 \varepsilon_0 + \alpha_1 G\varepsilon_0 + \cdots + \alpha_k G^k\varepsilon_0 \triangleq p_k(G)\varepsilon_0, \tag{32}$$

in which $p_k(G) = \sum_{i=0}^{k} \alpha_i G^i$ is a k-degree polynomial, and it satisfies $p_k(I) = I$. We look forward to choosing the appropriate coefficients $\alpha_i$ to let $\tilde{\mathbf{x}}^{(k)} - \mathbf{x}^*$ be as small as possible and increase the convergence rate. Obviously, this is an optimization problem. According to the definition in Equation (32), we can obtain

$$\left\| \tilde{\mathbf{x}}^{(k)} - \mathbf{x}^* \right\|_2 = \|p_k(G)\varepsilon_0\|_2 \leq \|p_k(G)\|_2 \cdot \|\varepsilon_0\|_2, \tag{33}$$

then, we need to solve the minimization problem below

$$\min_{p_k \in P_k, p_k(I) = I} \|p_k(G)\|_2, \tag{34}$$

in which $P_k$ is the set of all polynomials, where the capital $P$ stands for polynomials and the lowercase $k$ stands for the maximum degree. To solve the above problem, in particular, we assume that the iteration matrix $G$ is a symmetric matrix, which means that $G$ has the spectral decomposition as follows:

$$G = Q\Lambda Q^T, \tag{35}$$

where $\Lambda$ is a diagonal matrix and $Q$ is an orthogonal matrix; therefore,

$$
\begin{aligned}
\min_{p_k \in P_k, p_k(I)=I} \|p_k(G)\|_2 &= \min_{p_k \in P_k, p_k(I)=I} \|p_k(\Lambda)\|_2 \\
&= \min_{p_k \in P_k, p_k(I)=I} \max_{1 \leq i \leq n} \{|p_k(\lambda_i)|\} \\
&= \min_{p_k \in P_k, p_k(I)=I} \max_{\lambda \in [\lambda_n, \lambda_1]} \{|p_k(\lambda)|\},
\end{aligned}
\tag{36}
$$

where $\lambda_n$ and $\lambda_1$ are the maximum and the minimum eigenvalue, respectively. Next, we consider the iteration equation, Equation (10), in which the eigenvalues of the iteration matrix $G$ are all real numbers, the iterative matrix spectral radius $\rho = \rho(G) < 1$ and $\lambda \in [-\rho, \rho] \subset (-1, 1)$. Thus, Equation (36) can be converted into

$$
\min_{p_k \in P_k, p_k(I)=I} \max_{\lambda \in [-\rho, \rho]} \{|p_k(\lambda)|\}.
\tag{37}
$$

According to the above result, we can follow the Chebyshev polynomials' properties [31] to solve Equation (37) as follows:

$$
p_k(t) = \frac{T_k(t/\rho)}{T_k(1/\rho)},
\tag{38}
$$

where $T_k(t)$ is the k-degree Chebyshev polynomials. In fact, we can use the recurrence relation of the Chebyshev polynomials as follows:

$$
T_k(t) = 2t T_{k-1}(t) - T_{k-2}(t), \qquad k = 2, 3, \dots,
\tag{39}
$$

in addition, let $\mu_k = \frac{1}{T_k(1/\rho)}$ or $(T_k(1/\rho) = \frac{1}{\mu_k})$, and we can obtain a new recurrence relation equation as

$$
\frac{1}{\mu_k} = \frac{2}{\rho} \cdot \frac{1}{\mu_{k-1}} - \frac{1}{\mu_{k-2}}.
\tag{40}
$$

Substituting Equation (40) into Equation (32), we can obtain

$$
\tilde{\mathbf{x}}^{(k)} = \frac{2\mu_k}{\mu_{(k-1)}} \cdot \frac{G}{\rho} \tilde{\mathbf{x}}^{(k-1)} - \frac{\mu_k}{\mu_{(k-2)}} \tilde{\mathbf{x}}^{(k-2)} + \frac{2\mu_k}{\mu_{(k-1)}\rho} \mathbf{d}.
\tag{41}
$$

### 3.2. Proposed CAOR Method

In light of the previous subsections, to improve the BER performance and alleviate the significant complexity, we joined the AOR iterative algorithm and the recursive characteristics of the Chebyshev polynomials to construct a two-stage receiver, as depicted in Figure 1, in which the second block has more apparently depicted the Chebyshev recursive procedure. It can speed up convergence with fewer iterations and produce better estimation results. Additionally, we call this proposed scheme Chebyshev-accelerated over-relaxation (CAOR) detection.

The first stage of the receiver is an initial phase, which consists of an accelerated over-relaxation (AOR)-based estimator and is intended to yield a rough initial estimate of the relaxation factor $\omega$, the acceleration parameter $\gamma$, the spectral radius of $G_{\gamma,\omega}$, and transmitted symbols. The second stage is a refined phase, the Chebyshev acceleration method is used for detection, and a more precise signal is produced through efficient iterative estimation. The procedure of the CAOR detection is shown in Algorithm 1.

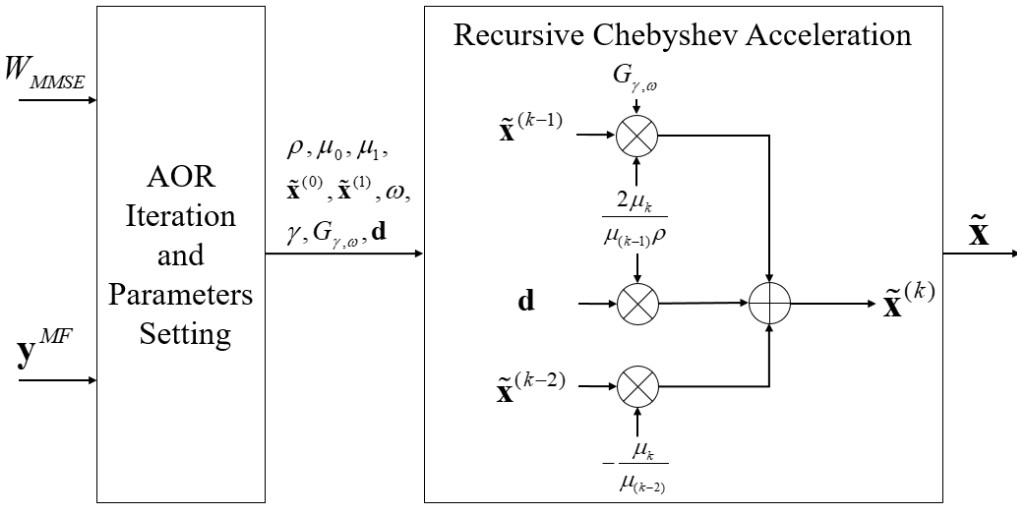

**Figure 1.** The block diagram of the proposed two-stage detection scheme.

---

**Algorithm 1** Chebyshev-Accelerated Over-Relaxation

**Receiver signal input :**

1.  $W_{MMSE} = H^H H + \sigma^2 I_{N_T} \triangleq A$, and $A = D + L + L^H$
2.  $\mathbf{y}^{MF} = H^H \mathbf{y} \triangleq \mathbf{b}$

**The first stage : (Initialize phase)**

1.  $G_{\gamma,\omega} = (D + \gamma L)^{-1}[(1 - \omega)D - (\omega - \gamma)L - \omega L^H]$
2.  $\mathbf{d} = \omega(D + \gamma L)^{-1}\mathbf{b}$
3.  **set** $\rho = \rho(G_{0,1})$, $\rho(G_{\gamma,\omega})$ is the spectral radius of $G_{\gamma,\omega}$
4.  **set** $\mu_0 = 1$, $\mu_1 = \rho$, $\tilde{\mathbf{x}}^{(0)} = \mathbf{x}^{(0)} = \mathbf{1}$ and $k = 1$

    **Compute** $\omega = \dfrac{1}{\sqrt{1-\mu_1^2}}, \gamma = \dfrac{2}{1+\sqrt{1-\mu_1^2}}$

    **Compute** $\mathbf{x}^{(1)} = G_{\gamma,\omega}\mathbf{x}^{(0)} + \mathbf{d} \triangleq \tilde{\mathbf{x}}^{(1)}$

**The second stage : (Refined phase)**

**While** not converge **do**

1.  $k = k + 1$
2.  $\mu_k = \left(\dfrac{2}{\rho} \cdot \dfrac{1}{\mu_{k-1}} - \dfrac{1}{\mu_{k-2}}\right)^{-1}$
3.  $\tilde{\mathbf{x}}^{(k)} = \dfrac{2\mu_k}{\mu_{(k-1)}} \cdot \dfrac{G_{\gamma,\omega}}{\rho}\tilde{\mathbf{x}}^{(k-1)} - \dfrac{\mu_k}{\mu_{(k-2)}}\tilde{\mathbf{x}}^{(k-2)} + \dfrac{2\mu_k}{\mu_{(k-1)}\rho}\mathbf{d}$

**end**
**set** $\tilde{\mathbf{x}} = \tilde{\mathbf{x}}^{(k)}$
**Receiver signal output :** The estimate of the transmitted signal vector $\tilde{\mathbf{x}}$

---

## 4. Simulation Results and Complexity Analysis

### 4.1. Simulation Results and Discussion

Some numerical simulations are conducted to evaluate the performance of the proposed two-stage receiver. Consider an $N_R \times N_T$ uplink MU M-MIMO OFDMA systems as depicted in Section 2, where $N_R$ and $N_T$ are the numbers of the antennas at the base station side and total user side, respectively. As in Table 1, simulation scenarios are 1024-quadrature amplitude modulation (QAM) and the number of subcarriers as 256, 100 OFDM symbols, cyclic prefix (CP) length is 64, each symbol has pilots as 20. In addition, assume that the channels are two-ray flat Rayleigh fading with additive white Gaussian noise (AWGN) and channel state information (CSI) is available at the receiver by LS estimation. Five receivers, including the SOR [27], the modified SOR [28], the SSOR [29], the AOR [30], and the proposed two-stage receiver, are conducted for comparison in terms of the bit error rate (BER) performance. Moreover, to accentuate the novelty of our proposed scheme, we

briefly describe the characteristics of previously published work and our proposed scheme in Table 2.

**Table 1.** Simulation scenarios.

| System | Parameter |
|---|---|
| Number of data subcarriers | 256 |
| Number of OFDM symbols | 100 |
| Modulation scheme | 1024-QAM |
| CP Length | 64 |
| Number of pilot data in one OFDM symbol | 20 |
| The maximum SNR | 60 |
| Channel | Rayleigh fading channel |
| Number of channel taps | 2 |
| Noise | AWGN |
| Channel estimation | LS |
| Monte Carlo (times) | 10,000 |

**Table 2.** A brief comparison of our proposed scheme and previously published work.

| Scheme | Brief Description |
|---|---|
| SOR [27] | SOR is a method of solving a linear system of equations derived by extrapolating the Gauss–Seidel method. |
| modified SOR [28] | Modified SOR is a variant of SOR, which changes certain parameters in the SOR algorithm and is nearly unaffected by the relaxation factor $\omega$ in lower modulation order. |
| SSOR [29] | SSOR combines two SOR sweeps in such a way that the resulting iteration matrix is similar to a symmetric matrix. |
| AOR [30] | The AOR iterative algorithm can be regarded as an extension of the SOR iterative algorithm, which is a two-parameter generalization of the SOR method. |
| CAOR | The CAOR method combines the AOR iterative algorithm and the recursive characteristics of the Chebyshev polynomials. |

First, we use Equations (16) and (17) to gain the appropriate relaxation parameter $\omega$ and the acceleration parameter $\gamma$, respectively, within the first stage. Apart from this, we show the BER performance of the proposed CAOR scheme against $\gamma$ and $\omega$ for different relaxation parameter $\omega$ values and acceleration parameter $\gamma$ values in Figures 2 and 3, respectively, when the number of iterations $i$ is 4, $N_R \times N_T = 64 \times 16$ and the SNR level is set to 40 dB. We can observe that if $\omega$ approaches 1.1 or so, the BER performance will improve; otherwise, BER performance will decrease. Thus, we believe that the choice of $\omega$ close to 1.1 and $\gamma$ close to 1.0 will yield the best parameter estimate. Simultaneously, the inferences from Figures 2 and 3 are consistent with the calculated values of Equations (16) and (17), respectively.

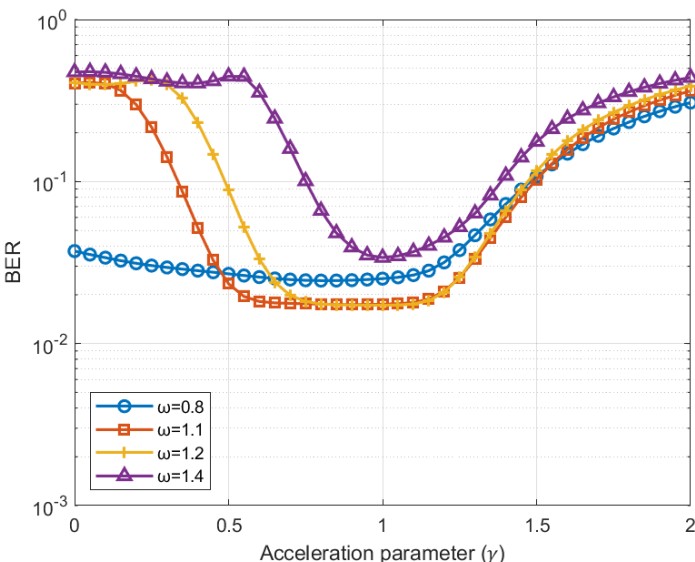

**Figure 2.** BER performance of CAOR method against $\gamma$ with SNR = 40 dB, $N_R \times N_T = 64 \times 16$, and number of iterations $i = 4$.

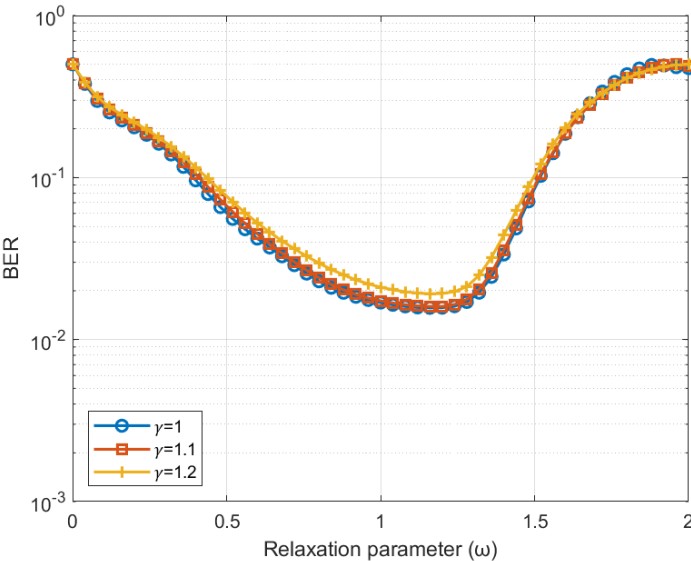

**Figure 3.** BER performance of CAOR method against $\omega$ with SNR = 40 dB, $N_R \times N_T = 64 \times 16$, and number of iterations $i = 4$.

Figures 4–6 depict the BER performance comparison with five different detection schemes, and $N_R$ is 64, 128, and 256, respectively. In Figure 4, we can roughly observe that the CAOR method is closest to the performance of conventional MMSE when the iteration number is equal to five. Moreover, when SNR = 45 dB, the BER performance of CAOR and AOR is approximately $0.6 \times 10^{-3}$ and $0.8 \times 10^{-3}$, respectively. In other words, compared to AOR, CAOR can improve BER performance by 25%.

When increasing $N_R$ to 128 and the number of iterations $i$ is 3, shown in Figure 5, we continued to observe that the BER performance of the proposed method almost overlaps with the conventional MMSE scheme, only requiring three iterations. Moreover, Figure 6 shows that when $N_R$ increases to 256 and is iterated twice only, we can note that the proposed method has the best BER performance in comparison with other detectors.

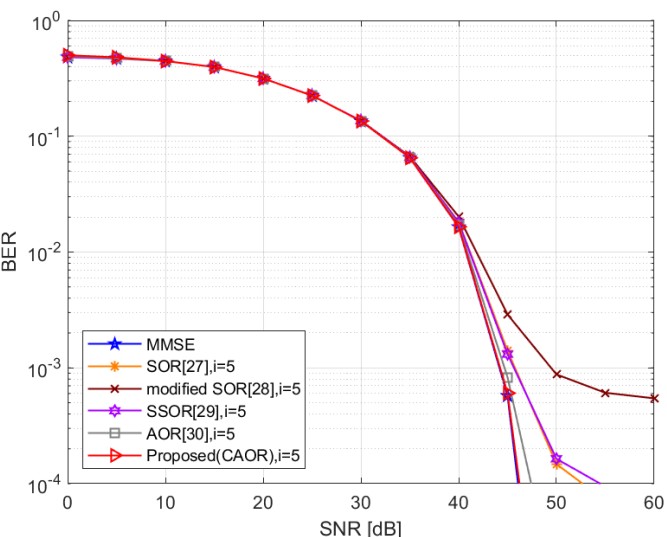

**Figure 4.** BER performance comparison for different detection schemes, $N_R \times N_T = 64 \times 16$.

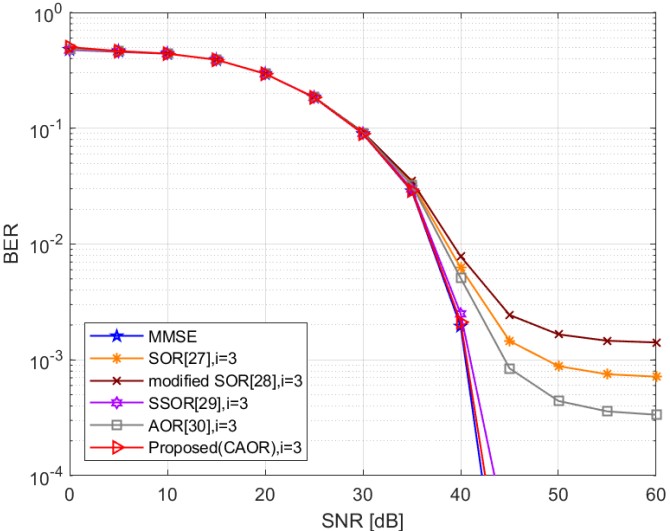

**Figure 5.** BER performance comparison for different detection schemes, $N_R \times N_T = 128 \times 16$.

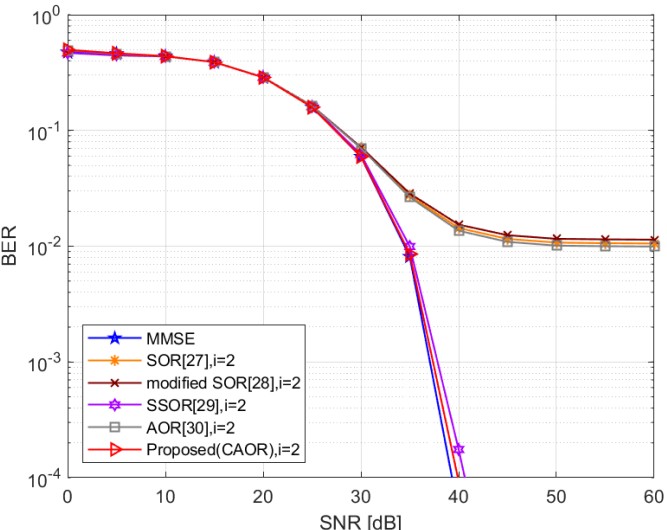

**Figure 6.** BER performance comparison for different detection schemes, $N_R \times N_T = 256 \times 16$.

We turn now to the ratio ($\beta$) analysis, which is the ratio between the number of base station antennas ($N_R$) and the total number of user antennas ($N_T$) [24]. As shown in Figure 7, the BER performance changes in terms of the $\beta$ of each detection scheme. Alongside this, when $\beta$ equals 16 and SNR is 40 dB, the BER performance of SSOR and CAOR is equal to $1.75 \times 10^{-4}$ and $9.27 \times 10^{-5}$, respectively. Taking it for granted, relative to the higher $\beta$ value, the BER performance of all detectors is improved due to the spatial diversity gain of a large number of antennas without exception. Moreover, we know that, regardless of the $\beta$ value, for the BER performance, SSOR is satisfactory, as is AOR, but our proposed scheme CAOR surpasses them all.

It is worth mentioning that when the number of antennas continues to increase, the number of iterations required is relatively reduced; simultaneously, the BER performance of our proposed method improves more definitely. On the whole, this is a good result for a 5G system with massive antennas.

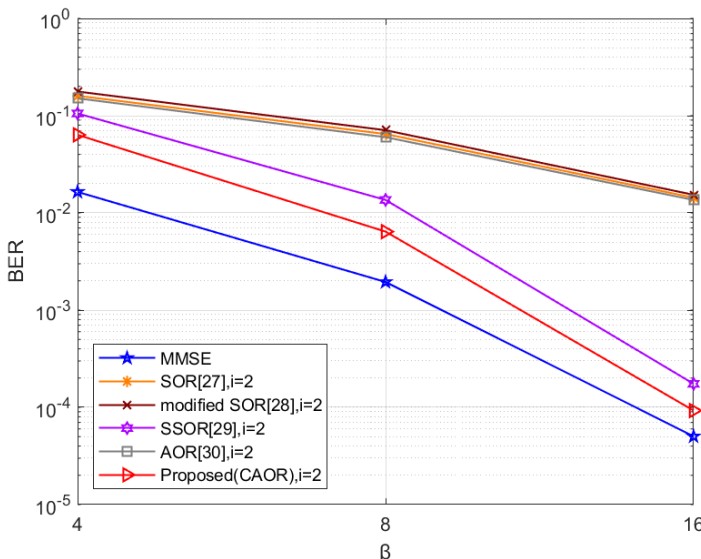

**Figure 7.** BER vs. $\beta$ for different detection schemes when $i = 2$ and SNR is 40 dB.

Moreover, in Table 3, we compare the degree of the BER performance approaching conventional MMSE among all detectors in terms of dB value (i.e., to obtain a more demarcated numerical comparison, we have taken $10log(\cdot)$ for the BER distance between the individual detector and conventional MMSE) under different $\beta$, where the CAOR method has the smallest distance with conventional MMSE. Apart from this, when $\beta$ is 16, the BER distance approach of CAOR to conventional MMSE is a negative dB value, which means that the BER distance between CAOR and conventional MMSE was drastically reduced and approached the optimal result. Of course, the smaller the dB value, the better and the closer to the optimum.

**Table 3.** Comparison of the degree to which the BER performance approaches that of conventional MMSE among all detectors in terms of dB value under different $\beta$ as $i = 2$ and SNR at 40 dB.

| Scheme | $\beta = 4$ | $\beta = 8$ | $\beta = 16$ |
|---|---|---|---|
| SOR | 9.43 dB | 15.19 dB | 24.52dB |
| modified SOR | 9.91 dB | 15.60 dB | 24.82 dB |
| SSOR | 7.34 dB | 7.89 dB | 3.93 dB |
| AOR | 9.13 dB | 14.86 dB | 24.30 dB |
| CAOR | 4.58 dB | 3.74 dB | $-0.738$ dB |

To understand the convergence of various detectors under different numbers of antennas, Figures 8–10 show the relationship between the number of iterations and the BER performance. The following samples are taken from them for observation and discussion.

In Figure 8, with five iterations, the BER performance of CAOR has a 95.52% improvement in BER compared with the conventional AOR. Similarly, when increasing $N_R$ to 128, with SNR at 45 dB, and the number of iteration equals 4, which is depicted in Figure 9, the BER performance of CAOR has a 47.77% improvement in BER compared with the conventional AOR. Additionally, Figure 10 shows the BER performance of CAOR under $N_R \times N_T = 256 \times 16$, SNR at 40 dB, and with two iterations, which has a 99.3% improvement in BER compared with the conventional AOR. As discussed above, besides the noise immunity of the massive-MIMO system being commendable, our proposed CAOR scheme has the fastest decline rate of BER and the additional merit of accelerating convergence.

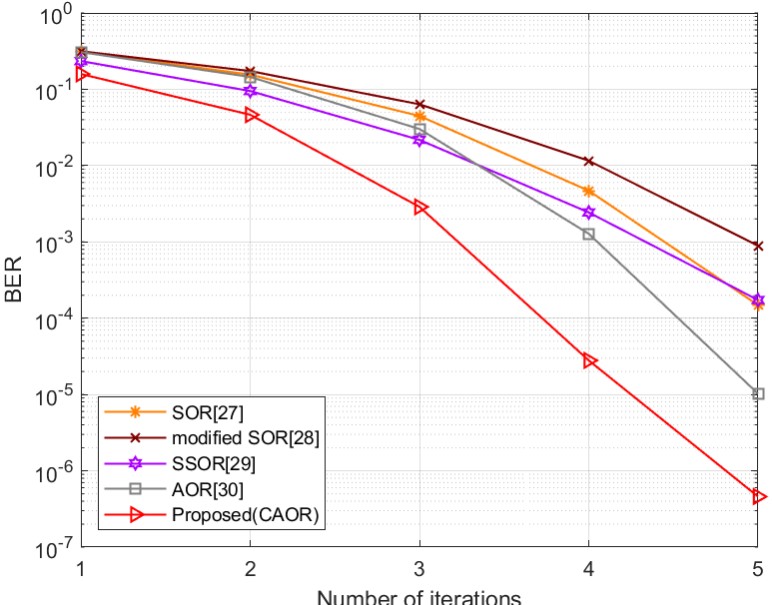

**Figure 8.** BER performance vs. number of iterations with $N_R \times N_T = 64 \times 16$ and SNR $= 50$ dB.

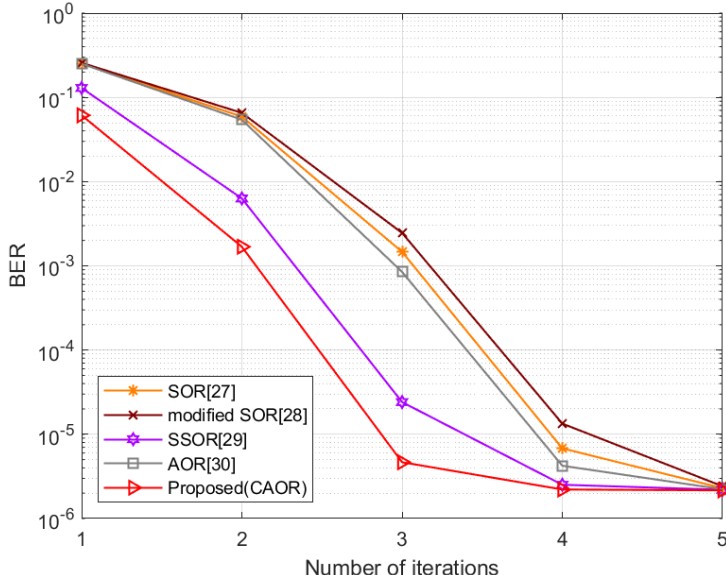

**Figure 9.** BER performance vs. number of iterations with $N_R \times N_T = 128 \times 16$ and SNR $= 45$ dB.

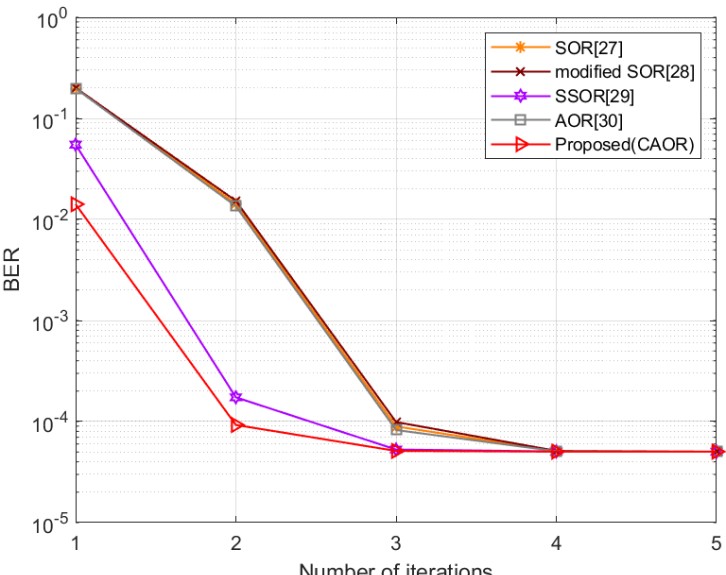

**Figure 10.** BER performance vs. number of iterations with $N_R \times N_T = 256 \times 16$ and SNR = 40 dB.

Next, in Table 4, we also compare the degree to which the BER performance approaches conventional MMSE among all detectors in terms of the dB value under different iteration numbers when $N_R \times N_T = 128 \times 16$ and SNR is 45 dB. Similarly to Table 3, to obtain a more demarcated numerical comparison, we have also taken $10log(\cdot)$ for the BER distance between the individual detector and conventional MMSE. Moreover, this table demonstrates that our proposed CAOR scheme has the smallest dB value, which means that the CAOR's BER performance is the closest to the conventional MMSE scheme regardless of the iteration number. Furthermore, we show the performance improvement of various receivers with the increase in antennas in Table 5, which refers to Figure 7 at the same time.

**Table 4.** Comparison of the degree to which the BER performance approaches that of conventional MMSE among all detectors in terms of dB value under different iteration numbers when $N_R \times N_T = 128 \times 16$ and SNR is 45 dB.

| Scheme | $i = 2$ | $i = 3$ | $i = 4$ |
|---|---|---|---|
| SOR | 44.35 dB | 28.40 dB | 3.31 dB |
| modified SOR | 44.80 dB | 30.62 dB | 7.14 dB |
| SSOR | 34.63 dB | 10.05 dB | −7.90 dB |
| AOR | 43.98 dB | 25.91 dB | −0.21 dB |
| CAOR | 28.94 dB | 0.61 dB | −16.88 dB |

**Table 5.** BER improvement rate of different detection schemes when $i = 2$ and SNR = 40 dB for $N_R = 128$ and $N_R = 256$. (Compared with $N_R = 64$).

| Scheme | $N_R = 128$ | $N_R = 256$ |
|---|---|---|
| SOR | 59.64% | 91.08% |
| modified SOR | 59.97% | 91.36% |
| SSOR | 87.08% | 99.83% |
| AOR | 60.12% | 90.98% |
| CAOR | 89.92% | 99.85% |

## 4.2. Computational Complexity Analysis

In this subsection, we evaluate the proposed detectors in terms of the computational complexity based on the number of complex multiplications required compared with other representative counterparts [27–30]. As in Table 6, we give the computational complexity

of different detection schemes, in which $i$ and $N_T$ denote the number of iterations and the total number of user antennas, respectively. In addition, the inverse matrix of each iteration method requires $2N_T^2 - N_T$ complex multiplications and additions (CMAs) [28]. We consider the CAOR iterative equation as follows:

$$\tilde{\mathbf{x}}^{(k)} = M^{-1} \cdot \frac{2\mu_k}{\mu_{(k-1)}\rho} \cdot (N\tilde{\mathbf{x}}^{(k-1)} + \omega\mathbf{b}) - \frac{\mu_k}{\mu_{(k-2)}}\tilde{\mathbf{x}}^{(k-2)}, \qquad (42)$$

in which $M^{-1} = (D + \gamma L)^{-1}$ and $N = (1 - \omega)D - (\omega - \gamma)L - \omega L^H$ require $2N_T^2 - N_T$ CMAs and $N_T^2$ CMAs, respectively. Additionally, $\omega\mathbf{b}$ requires $N_T$ CMAs; as discussed above, we can obtain $3N_T^2$ CMAs, which is part of the CAOR's computational complexity, as shown in Table 6. For the iteration part of CAOR, $M^{-1} \cdot \frac{2\mu_k}{\mu_{(k-1)}\rho} \cdot (N\tilde{\mathbf{x}}^{(k-1)} + \omega\mathbf{b})$ and $-\frac{\mu_k}{\mu_{(k-2)}}\tilde{\mathbf{x}}^{(k-2)}$ require $i(3N_T^2 + N_T)$ CMAs and $2iN_T$ CMAs, respectively. Thus, the iteration part of the CAOR's computational complexity requires $3i(N_T^2 + N_T)$ CMAs altogether. Further, the numerical comparison of the complexity is shown in Table 7.

**Table 6.** Computational complexity of different detection schemes.

| Detection Scheme | Complex Multiplications and Additions (CMAs) |
|---|---|
| SOR | $\frac{1}{2}(5N_T^2 + N_T) + i(2N_T^2 + N_T)$ |
| modified SOR | $\frac{1}{2}(5N_T^2 - 3N_T) + i(2N_T^2 + N_T)$ |
| SSOR | $5N_T^2 - N_T + 2i(2N_T^2 + N_T)$ |
| AOR | $3N_T^2 + 3iN_T^2$ |
| CAOR | $3N_T^2 + 3i(N_T^2 + N_T)$ |

**Table 7.** The numerical comparison of the complexity for different detection schemes with $N_T = 16$.

| Detection Scheme | CMAs $i = 2$ | CMAs $i = 3$ | CMAs $i = 4$ | CMAs $i = 5$ |
|---|---|---|---|---|
| SOR | 1704 | 2232 | 2760 | 3288 |
| modified SOR | 1672 | 2200 | 2728 | 3256 |
| SSOR | 3376 | 4432 | 5488 | 6544 |
| AOR | 2304 | 3072 | 3840 | 4608 |
| CAOR | 2400 | 3216 | 4032 | 4848 |

Regarding the complexity and performance factors, as shown in Figure 5, even though the BER performance of SSOR [29] is approximately $2.5 \times 10^{-3}$, which is also close to the conventional MMSE scheme, its complexity is approximately 37.8% higher than the CAOR method, which can be verified by the equation within Table 6. Meanwhile, we also know that the computational complexity of SSOR is the highest among all detectors, which is shown in Table 7.

For a more detailed discussion, we first consider four iterations in Figure 8; the CAOR method has improvements of 93% in BER compared with the conventional AOR. Meanwhile, the complexity only increases by 192 CMAs (increases by 5%). On the other hand, the CAOR method with $i = 4$ only requires 4032 CMAs, which can achieve similar performance to the conventional AOR with $i = 5$ requiring 4608 CMAs. Alongside this, when $N_R \times N_T = 64 \times 16$ and the iteration number is four, SSOR's complexity and BER performance both are worse than AOR, not to mention compared with the CAOR method. The more complete numerical computational complexity in terms of iteration number for different detectors is shown in Table 7.

Secondly, to further observe the effect on BER and complexity when the number of antennas is increased, according to Figures 6 and 10, as $N_R$ increases, the number of iterations required apparently gradually decreases. Furthermore, the BER performance and the computational complexity of CAOR decrease more than other detection methods.

Clearly, as shown in Figure 10, the BER performance of the proposed scheme only requires two iterations due to the sufficient number of antennas, which can achieve better BER performance than SOR, modified SOR, SSOR, and AOR. Meanwhile, as shown in Table 7, the complexity of CAOR at $i = 2$ and AOR at $i = 3$; they require 2400 CMAs and 3072 CMAs, respectively. The BER performance of CAOR and AOR is equal to $9.26 \times 10^{-5}$ and $8.24 \times 10^{-5}$, respectively; they are very close to each other. In light of the above discussion, we can conclude that as the number of antennas increases, the number of iterations will decrease gradually, and thus it is conducive to reducing the CMAs.

Therefore, as a whole, the computational complexity of CAOR is actually lower than that of conventional AOR. To show the computational complexity of different detection schemes more clearly, we utilize a bar graph to depict them in Figure 11.

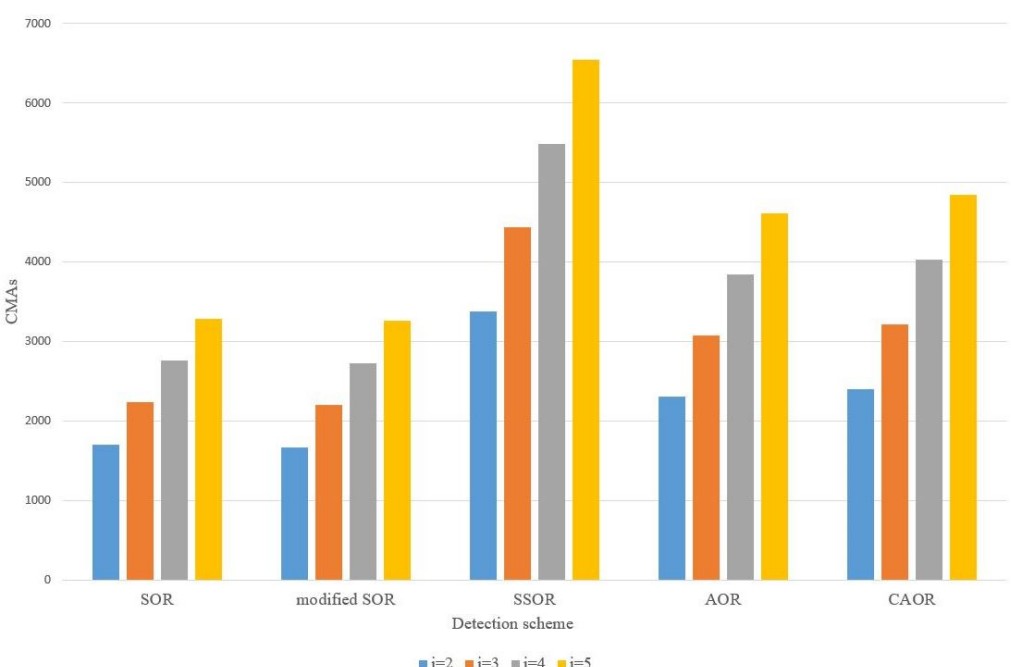

**Figure 11.** Computational complexity of different detection schemes with $N_T = 16$.

## 5. Conclusions

In this paper, a novel two-stage receiver combining Chebyshev acceleration and the AOR algorithm (CAOR) is proposed to improve the convergence rate and achieve remarkable BER performance in a small number of iterations. Numerical results demonstrate the performance comparison with the different detection methods, in which the performance of CAOR is the best among all other detection methods. Although this method raises the computational load slightly more than AOR, fortunately, this is acceptable since the hardware's computation ability is growing nowadays. Moreover, it can achieve a similar performance as SSOR and AOR with fewer iterations. Hence, the computational complexity of the proposed method is actually lower than expected. To sum up, regarding the BER performance and complexity, our proposed scheme is beyond comparison.

In summary, the M-MIMO has the ability to promote BER performance significantly in the 5G OFDMA systems, and our proposed method can achieve an appropriate acceleration parameter $\gamma$ and relaxation parameter $\omega$ and has the best BER under moderate computational complexity, so it is very suitable to apply to the current and next-generation wireless systems.

**Author Contributions:** Conceptualization, Y.-P.T., C.-Y.C. and K.-H.L.; methodology, Y.-P.T. and C.-Y.C.; software, C.-Y.C. and Y.-P.T.; validation, Y.-P.T., C.-Y.C. and K.-H.L.; formal analysis, Y.-P.T. and K.-H.L.; investigation, Y.-P.T. and C.-Y.C.; data curation, Y.-P.T. and C.-Y.C.; writing—original draft preparation, Y.-P.T., K.-H.L. and C.-Y.C.; writing—review and editing, Y.-P.T., C.-Y.C. and K.-H.L.; visualization, Y.-P.T. and K.-H.L.; supervision, Y.-P.T.; project administration, Y.-P.T. and K.-H.L.; funding acquisition, K.-H.L. and Y.-P.T. All authors have read and agreed to the published version of the manuscript.

**Funding:** This research received no external funding.

**Acknowledgments:** This research was supported in part by the Ministry of Science and Technology in Taiwan under the grant numbers MOST 109-2637-E-150-002 and MOST 110-2637-E-150-011.

**Conflicts of Interest:** The authors declare no conflicts of interest.

**Abbreviations**

The following abbreviations are used in this manuscript:

| | |
|---|---|
| AOR | Accelerated over-relaxation |
| AWGN | Additive white Gaussian noise |
| BER | Bit error rate |
| BS | Base station |
| CAOR | Chebyshev-accelerated over-relaxation |
| CMAs | Complex multiplications and additions |
| CP | Cyclic prefix |
| CSI | Channel state information |
| i.i.d. | Independent and identically distributed |
| IoT | Internet of Things |
| ISI | Inter-symbol interference |
| LS | Least square |
| M-MIMO | Massive multiple-input multiple-output |
| MIMO | Multiple-input multiple-output |
| MMSE | Minimum mean-squared error |
| MU | Multiuser |
| MUD | Multiuser detector |
| OFDM | Orthogonal frequency-division multiplexing |
| OFDMA | Orthogonal frequency-division multiple access |
| QAM | Quadrature amplitude modulation |
| SOR | Successive over-relaxation |
| SSOR | Symmetric successive over-relaxation |
| ZF | Zero-forcing |

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
