# Peer review of "An Efficient Two-Stage Receiver Base on AOR Iterative Algorithm and Chebyshev Acceleration for Uplink Multiuser Massive-MIMO OFDM Systems"

_electronics, doi:10.3390/electronics11010092_

Round 1
Reviewer 1 Report
The paper has an interesting subject. It proposes a novel two-stage receiver combining Chebyshev acceleration and
AOR algorithm used in Massive MIMO systems. The results showed that this receiver improves the convergence rate and achieve a remarkable BER performance in a small number of iterations. Its performances are better compared with the other existing detection methods.
The paper is well organized and it technically sounds.
The method is clearly presented. The paper has a good state-of-the-art.
There are 2 minor observations: some of the acronyms are not explained at the first use and on page 4 is "the the" after equation 6.
Author Response
Commends: There are 2 minor observations: some of the acronyms are not explained at the first use and on page 4 is "the the" after equation 6.
Response #1: Thanks for your kind comments and suggestions. We have carefully searched the problems that acronyms are not explained at the first within the paper. And have been finished explained. The revision is the following and we also highlight it in uploaded PDF file.
In line 8 of page 1: accelerated over-relaxation (AOR)
In line 13 of page 2: zero-forcing (ZF) and minimum mean-square error (MMSE)
In line 9 of page 3: orthogonal frequency-division multiple access (OFDMA)
Response #2: Thanks for your kind comment. We deleted the redundancy text “the”, which is behind equation 6 on page 4.

Reviewer 2 Report
The work presented within this paper is somehow good, but still need some improvements.
- A table of comparison with previous published work is needed. This will clearly claim the novelty of this present work.
- Paper should be revisited in terms of English.
- more close and relevant references may be added in the revised version.
Author Response
Point #1: A table of comparison with previous published work is needed. This will clearly claim the novelty of this present work.
Response #1: Thank you very much for your precious suggestion. To more clear the novelty of our proposed scheme, we have added a table of comparisons with previously published work to improve the readability of the paper. The revision is the following and we also highlight it in uploaded PDF file.
In line 13 of page 9:
Moreover, to accentuate the novelty of our proposed scheme, we briefly describe the characteristics of previously published work and our proposed scheme in Table 2.
Point #2: Paper should be revisited in terms of English.
Response #2: Thank you for your kind comments. We have further rewritten some sentences to improve the readability in terms of English and we also highlight it in uploaded PDF file..
Point #3: more close and relevant references may be added in the revised version.
Response #3: Thank you for your comments. To improve the readability of the paper, we have added some more close and relevant references in the revised version. The relative reference index order is also revised at the same time. The revision is the following and we also highlight it in uploaded PDF file.
- Boccardi, F.; Heath, Jr., R.W.; Lozano, A.; Marzetta, T.L.; Popovski, P. Five Disruptive Technology Directions for 5G. IEEE Commun. Mag. Feb. 2014, 52, pp. 74–80.
- Di Renzo, M.; Haas, H.; Ghrayeb, A.; Sugiura, S.; Hanzo, L. Spatial modulation for generalized MIMO: Challenges, opportunities, and implementation. IEEE Trans. Veh. Technol. 2014, 57, pp. 2228–2241.
- Larsson, E.G. MIMO Detection Methods: How They Work [lecture notes]. IEEE Signal Process. Mag. May 2009, 26, pp. 91–95.
- Morelli, M.; Kuo, C.-C. J.; Pun, M.-O. Synchronization Techniques for Orthogonal Frequency Division Multiple Access (OFDMA): A Tutorial Review. Proc. IEEE Jul. 2007, 95, pp. 1394–1427.
- Hadjidimos, A. Successive overrelaxation (SOR) and related methods. Computational and Applied Mathematics, Nov. 2000, 123, pp. 177–199.

Reviewer 3 Report
The paper deals with a novel two stage receiver architecture for the MU M-MIMO systems which aims to address trade-off between BER and complexity load. Though authors did not provide measurement results, the paper is well written and original. Simulation results are clearly presented and the overall comparison against state of the art as well as trade-off in terms of BER and complexity load of the architecture with respect to other works have been well outlined.
However, aiming at improve the readability of the paper, authors should redraw Fig. 1 since it is out of focus and with lower dpi than other figures. Furthermore, Tab. 3 has a footnote with “*” but such symbol does not refer to any element in particular, so authors should better clarify this choice in the text or in the caption.
Author Response
Point # 1: authors should redraw Fig. 1 since it is out of focus and with lower dpi than other figures.
Response #1: Thank you for your kind comment. To promote the focus of Chebyshev recursive procedure and increase picture clarity, Figure 1 has been redrawn and added text description. The revision is the following and we also highlight it in uploaded PDF file.
In line 22 of page 7:
In light of previous subsections, to prompt the BER performance and alleviate the significant complexity, we joined the AOR iterative algorithm and the recursive characteristics of the Chebyshev polynomials to construct a two-stage receiver as depicted in Figure 1, in which, the second block has more apparently depicted the Chebyshev recursive procedure.
Point # 2: Tab. 3 has a footnote with “*” but such symbol does not refer to any element in particular, so authors should better clarify this choice in the text or in the caption.
Response #2: Thank you so much for your kind comment. This is author oversight, and we have deleted the footnote. The revision is the following and we also highlight it in uploaded PDF file.
In line 3 of page 14:
Similar to Table 3, to obtain a more demarcated numerical comparison, we have also taken 10log(.) for the BER distance between the individual detector and conventional MMSE.
